# The Caribou (*Rangifer tarandus*) Genome

**DOI:** 10.3390/genes10070540

**Published:** 2019-07-17

**Authors:** Rebecca S. Taylor, Rebekah L. Horn, Xi Zhang, G. Brian Golding, Micheline Manseau, Paul J. Wilson

**Affiliations:** 1Biology Department, Trent University, 1600 West Bank Drive, Peterborough, ON K9J 7B8, Canada; 2Department of Biology, McMaster University, 1280 Main St. West, Hamilton, ON L8S 4K1, Canada; 3Science and Technology Division, Environment and Climate Change Canada, 1125 Colonel By Drive, Ottawa, ON K1S 5R1, Canada

**Keywords:** caribou, reindeer, *Rangifer tarandus*, genome, genome assembly

## Abstract

*Rangifer tarandus*, known as caribou or reindeer, is a widespread circumpolar species which presents significant variability in their morphology, ecology, and genetics. A genome was sequenced from a male boreal caribou (*R. t. caribou*) from Manitoba, Canada. Both paired end and Chicago libraries were constructed and sequenced on Illumina platforms. The final assembly consists of approximately 2.205 Gb, and has a scaffold N50 of 11.765 Mb. BUSCO (Benchmarking Universal Single-Copy Orthologs) reconstructed 3820 (93.1%) complete mammalian genes, and genome annotation identified the locations of 33,177 protein-coding genes. An alignment to the bovine genome was carried out, indicating sequence coverage on all bovine chromosomes. A high-quality reference genome will be invaluable for evolutionary research and for conservation efforts for the species. Further information about the genome, including a FASTA file of the assembly and the annotation files, is available on our caribou genome website. Raw sequence data is available at the National Centre for Biotechnology Information (NCBI), under the BioProject accession number PRJNA549927.

## 1. Introduction

*Rangifer tarandus*, known as caribou in North America and reindeer in Europe and Asia, is the most widespread circumpolar ungulate species [1]. The species occurs in a variety of ecozones, including High Arctic, taiga, mountains, and boreal, and as such are hugely variable in their morphology, ecology, and genetics [1]. In Canada, caribou are declining due to a number of stressors, including anthropogenic disturbances and climate changes [2,3]. Even though many caribou populations fluctuate over time, the current declines appear to be surpassing the ability of many herds to recover [3]. Consequently, caribou are a conservation concern in most of Canada [3,4].

Although recognized as one species across its vast circumpolar range, caribou has a complex history and existed in multiple refugia during the Pleistocene leading to three main lineages—a Beringian, a North American and a High Arctic lineage [5,6]. Currently, caribou are divided into multiple subspecies, the number of which has been disputed but with nine listed in Banfield’s often cited revision of their taxonomy [1]. Taxonomic clarification was provided by COSEWIC in 2011 [4] given the considerable variability even within some of the designated subspecies. In Canada, caribou are currently divided into 11 extant and 1 extinct Designatable Units (DUs) to ensure the conservation of all caribou diversity [4]; however, further research is essential to clarify the delineation and evolutionary history of caribou groups and to elucidate functional genomic regions underlying ecological adaptation. To help achieve this goal, we have sequenced a high-quality caribou reference genome from a male boreal caribou (*R. t. caribou*; COSEWIC DU6) from Snow Lake, Manitoba, Canada.

## 2. Materials and Methods

Neck muscle tissue was collected from an adult male boreal caribou (*R. t. caribou)*, which had been killed on a road in Manitoba in October 2009. The tissue was stored in RNA later ICE (Thermo Fisher Scientific, MA, USA). Phenol chloroform extraction [7] was performed using 0.2 g of tissue, and eluted in Tris-ethylenediaminetetraacetic acid (TE) buffer at 100 µL. The DNA was shipped to Dovetail Genomics for library preparation, sequencing and assembly.

Three Chicago libraries were prepared as described previously elsewhere [8]. Briefly, for each library, ~500 ng of high molecular weight genomic DNA (gDNA; mean fragment length = 85 kb) was reconstituted into chromatin in vitro and fixed with formaldehyde. Fixed chromatin was digested with DpnII, the 5’ overhangs filled in with biotinylated nucleotides, and then free blunt ends were ligated. After ligation, crosslinks were reversed and the DNA purified from protein. Purified DNA was treated to remove biotin that was not internal to ligated fragments. The DNA was then sheared to ~350 bp mean fragment size and sequencing libraries were generated using NEBNext Ultra enzymes and Illumina-compatible adapters (New England BioLabs, Ipswich, MA, USA). Biotin-containing fragments were isolated using streptavidin beads before PCR enrichment of each library. The libraries were sequenced on an Illumina HiSeq X (Illumina, San Diego, CA, USA). The number and length of read pairs produced for each library was: 123 million, 2 × 101 bp for library 1; 66 million, 2 × 101 bp for library 2; and 125 million, 2 × 101 bp for library 3. Together, these Chicago library reads provided 50.8 × physical coverage of the genome (1–50 kb pairs).

A de novo assembly was constructed using a combination of paired end reads (mean insert sizes ~350 bp and 550 bp), which were sequenced on an Illumina HiSeq2500 (Illumina, San Diego, CA, USA) and an Illumina HiSeq X (Illumina, San Diego, CA, USA), respectively. De novo assembly was performed using Meraculous (version 2.2.2.5) [9] with a *kmer* (*k*) size of 43. The input data consisted 1.51 billion read pairs sequenced from paired end libraries (totaling 453 Gbp). Reads were trimmed for quality, sequencing adapters, and mate pair adapters using Trimmomatic [10].

The input de novo assembly, shotgun reads, and Chicago library reads were used as input data for HiRise, a software pipeline designed specifically for using proximity ligation data to scaffold genome assemblies [8]. Shotgun and Chicago library sequences were aligned to the draft input assembly using a modified SNAP read mapper (http://snap.cs.berkeley.edu). The separations of Chicago read pairs mapped within draft scaffolds were analyzed by HiRise to produce a likelihood model for genomic distance between read pairs, and the model was used to identify and break putative misjoins, to score prospective joins, and make joins above a threshold. After scaffolding, shotgun sequences were used to close gaps between contigs. Raw sequence data is available at the National Centre for Biotechnology Information (NCBI), under the BioProject accession number PRJNA549927. Mitochondrial DNA is not included in the assembly as a full mitogenome assessment is currently underway with additional samples and will be released in the future.

We used the gene prediction program AUGUSTUS 2.5.5 [11] to annotate the genome using predictions based on human genes. The genome was masked using RepeatMasker 3.2.6 [12] and run in Augustus using a partial gene model allowing the prediction of incomplete genes at the sequence boundaries.

We ran the final genome FASTA file through the stats.sh function of BBMap 38.42 [13] to calculate genome statistics such as the N50. We also used BUSCO (Benchmarking Universal Single-Copy Orthologs; [14]) to reconstruct 4104 conserved mammalian genes to assess genome completeness. We aligned the caribou genome to the bovine reference genome, as it is the most closely related (estimated to share a common ancestor around 25.8 million years ago [15]), highest quality reference genome. We downloaded the FASTA sequence from the bovine genome database (bovinegenome.org [16]), and aligned it to our genome using BWA-MEM [17]. Using the alignment, we created a Jupiter plot using the script written by J. Chu [18] and Circos 0.69-3 [19]. We plotted the largest scaffolds covering 75% of the bovine genome (as the figure becomes crowded and unclear when showing more) to assess synteny between the two genomes. We also downloaded the consensus FASTA sequence for a previously published reindeer genome from Inner Mongolia [20] and did an alignment in the same way with the bovine genome. We used QUAST 5.0.2 [21] to assess the quality of both our caribou assembly and the reindeer assembly in comparison to the bovine genome.

## 3. Results and Discussion

The final *Rangifer tarandus* genome assembly consists of approximately 2.205 Gb, with a scaffold N50 of 11,765,000 base pairs (Table 1), and a GC content of 41.44% (Table 2). AUGUSTUS identified the locations of 33,177 protein-coding genes, and BUSCO indicated the presence of 3820 (93.1%) complete mammalian genes of the 4104 searched for. Our quality assessment statistics are similar to those of other recent non-model mammal species genome assemblies, including the American brown bear (*Ursos arctos* ssp. *horribilis*) [22]; the beluga whale (*Delphinapterus leucas*) [23]; and the northern sea otter (*Enhydra lutris kenyoni*) [24]. The previously published *Rangifer tarandus* genome, sequenced from a domesticated individual from Inner Mongolia [20], consists of 58,765 scaffolds, with a scaffold N50 of 986, 392 bp, and successfully reconstructed 92.6% of the BUSCO genes, indicating our genome to be a more contiguous assembly. We used the Chicago method which produces proximity ligation libraries that have a relationship between within-read pair distance and read count. This produces long-range sequence scaffolds during the assembly of genomes [8], and increased our scaffold contiguity compared to the reindeer.

The reindeer genome is larger, at 2.64 Gb, and so may cover more of the genome in total. However, QUAST results indicated that the missing data is much higher for the reindeer genome, with 3.6% of their bases as N’s, whereas our assembly consists of 0.7% N’s. Similarly, the reindeer annotation recovered fewer genes than we did [20], which may be because our annotation does not account for pseudogenes or incomplete proteins. However, it could also be because their assembly is fragmented with a higher percentage of missing data, which may have impacted the detection of genes. Using QUAST, our caribou assembly recovered 8402 genes from the Bovine annotation, whereas the reindeer recovered 5755. In addition, we recovered more conserved genes during the BUSCO analysis, suggesting the difference in the number of genes recovered during genome annotation is related to the differences in genome contiguity.

Our genome sequence is the first North American *Rangifer tarandus* (caribou) genome, and is from a wild animal which is important as genetic differences have been found between domesticated and wild reindeer [25]. Therefore, both genomes likely represent important but different genomic variation. As one of our primary aims is to use the genome to aid with the conservation of wild populations, our assembly represents a valuable resource.

The Jupiter plot displays the largest 312 caribou scaffolds, out of a total of 4699, which cover 75% of the bovine genome (Figure 1). The coloured bands represent synteny between the caribou and bovine assemblies. The lines crossing the circle could be genomic rearrangements, but also likely represent break points in the assembly, particularly when appearing at the edges of scaffolds. The BWA results indicated that the sex chromosomes appear on smaller scaffolds within the caribou assembly, explaining why there are few alignments showing in the Jupiter plot. This reflects the difficulty in assembling large contigs and scaffolds for the sex chromosomes due to their highly repetitive nature [26]. Overall, the BWA alignment and Jupiter plot show good synteny between our caribou assembly and the bovine reference genome, and tells us on which bovine chromosomes the caribou scaffolds are syntetic to (see Appendix A for a list of which bovine chromosome the caribou scaffolds align to, and the caribou genome website for BAM and BED alignment files for our alignment to the bovine genome). The Jupiter alignment of the reindeer to the bovine genome also showed our assembly to be more contiguous in comparison (Appendix A).

Information about the genome is available and continuously updated at www.caribougenome.ca. The website includes a BLAST function, as well as the ability to download the genome in FASTA format, the annotation (gff3 and bed) files, and a RepeatMasked version of the genome. The availability of a high-quality genome will be invaluable for answering evolutionary questions relating to this wide ranging and variable species, but also for conservation efforts. For example, a larger number of molecular markers can be developed using the whole genome data, as well as investigation into variation of potentially functional importance [27].

## Figures and Tables

**Figure 1 genes-10-00540-f001:**
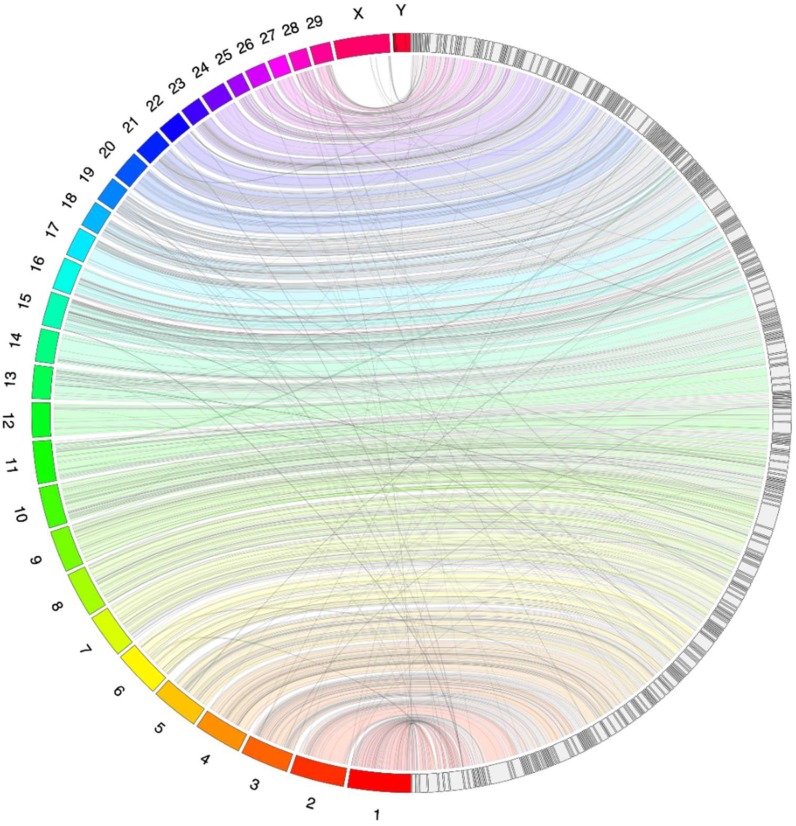
A Jupiter plot showing an alignment between the bovine chromosomes and the caribou genome assembly. The left of the circle shows the numbered bovine chromosomes, and the right of the circle has the largest 312 scaffolds from our assembly, which cover 75% of the bovine genome. Coloured bands represent synteny between the genomes, and lines crossing the circle indicate genomic rearrangements, or break points in the scaffolds.

**Table 1 genes-10-00540-t001:** Assembly statistics of the caribou genome.

Statistic	*Rangifer tarandus* genome
Scaffold sequence total (bp)	22.052 × 10^8^
Number scaffolds	4699
Scaffold N50 (bp)	11.765 × 10^6^
Scaffold L50	52
Scaffold N90 (bp)	89.704 × 10^4^
Scaffold L90	289
Contig sequence total (bp)	21.893 × 10^8^
Number contigs	146,562
Contig N50 (bp)	32.819 × 10^3^
Contig L50	19,701
Contig N90 (bp)	89.140 × 10^2^
Contig L90	68,199

**Table 2 genes-10-00540-t002:** Nucleotide base composition of the caribou genome assembly statistics of the caribou genome.

A	C	G	T	N
29.27%	20.72%	20.73%	29.28%	0.72%

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
