# Peer review of "The Caribou (Rangifer tarandus) Genome"

_genes, 2019, doi:10.3390/genes10070540_

Reviewer 1 Report

Minor comments:

As far as I can tell no explicit justification for comparing synteny to cows is provided, although the explanation is fairly obvious. But, to help everyone to understand your rational you could a) indicate that they are the most closely related highest quality reference and b) provide some sentence about the time since divergence to place the expected synteny into some context. 

As everyone knows, a large N50 by itself is not an indicator of a good assembly (which is why you provide the synteny comparison to cows), so it would be informative to have a synteny comparison to the previously sequenced Rangifer tarandus. Of course, if the assemblies are largely syntenic I would have no qualms with calling the new assembly more complete, but the comparison may not provide much additional information for the additional effort, so it is by no means necessary. However, you may want to provide some short justification for why such a comparison was not provided. 

Author Response

As far as I can tell no explicit justification for comparing synteny to cows is provided, although the explanation is fairly obvious. But, to help everyone to understand your rational you could a) indicate that they are the most closely related highest quality reference and b) provide some sentence about the time since divergence to place the expected synteny into some context. 

-  We have added a comment into the text to justify the comparison, and have added a citation to state the estimated time since their common ancestor.

As everyone knows, a large N50 by itself is not an indicator of a good assembly (which is why you provide the synteny comparison to cows), so it would be informative to have a synteny comparison to the previously sequenced Rangifer tarandus. Of course, if the assemblies are largely syntenic I would have no qualms with calling the new assembly more complete, but the comparison may not provide much additional information for the additional effort, so it is by no means necessary. However, you may want to provide some short justification for why such a comparison was not provided. 

- We have done an alignment of the reindeer to the high-quality bovine genome to compare with ours, and included the figure in the supplementary material. Comparison between the figures gives a clear visualisation of the fact that our reference genome is much more contiguous than the reindeer assembly. We have included our alignment in the main text and the reindeer alignment in the supplementary, although are happy to include either figure as supplementary or in the main text.

Reviewer 2 Report

I have no comments for the authors.

Author Response

I have no comments for the authors.

- We are glad you like our manuscript.

Reviewer 3 Report

This is a descriptive manuscript focused on denovo assembled genome of the wild Caribou. The authors use a combination of short-read sequencing with Chicago libraries for linkage to create a draft assembly which is 2.2 Gbp long and consists of 4,699 scaffolds. The description of the methods is sufficient, and the assembled genome and the identified genes will be a resource to the community. However, the analyses of the genome are minimal, and leave a lot to be desired. I understand the argument that this manuscript is not intended to be focused on evolutionary  themes, so I will limit my suggestions (for the most part) to analyses that are either focused on the assembly or the use of this genome as a resource:

1. I would suggest the authors run RepeatMasker to catalog the repeats in their genome. This is a relatively straightforwards analysis, but the repeatmasked genome will facilitate several downstream analyses, adding value to this resource.

2. A genome of the reindeer has been published (https://doi.org/10.1093/gigascience/gix102) earlier, as the authors mention in their report. That genome is 2.64 Gbp in length compared to 2.2 Gbp as assembled by the authors. kmer analyses show (as per Li et al.) that the expected genome size is 2.76 Gbp. I would suggest that the authors compare their genome to the published reindeer genome and comment on the differences in the assembled genome size for the two samples.

3. The authors mention that their assembled genome is more contiguous compared to the earlier published genome. Can they comment on whether the Chicago libraries are the reason for that, or is it that their assemblers do a better job compared to SOAPdenovo which was used in the other assembly?

4. Figure 1 does not add any value to the manuscript in my opinion. I understand that comparison to other assembled genomes should be done, but the description of the analysis is sufficient. The figure should be removed.

5. On the other hand, I do not understand the Supplementary file which is provided. The information I would like to see is the intervals in the assembled genome along with the corresponding coordinates in the bovine genome that they align to. I would suggest using a BEDPE sort of format which will be useful.

6. Another relatively straightforward analysis that seems to be missing is the analysis of the mitochondrion. Does the assembly include the mitochondrion. Li et al. used the reindeer genes to calculate the phylogenetic relationships to the other genomes. Is that relationship supported by the mtDNA analysis? 

7. How many of the genes identified by the authors are protein-coding. And can they enumerate the other classes of genes? How many rRNA, miRNA, tRNA genes were identified?

8. Kharzinova et al. (https://journals.plos.org/plosone/article?id=10.1371/journal.pone.0207944) published an analysis of the genetic diversity and population structure of domestic and wild reindeer. If you project the SNPs of the individual used in this analysis on to the MDS plot of the samples used in Kharzinova et al., how does it compare?

9. The authors mention that the site www.caribougenome.ca would go live by the time of the publication. Considering that this manuscript is primarily describing a resource, I would suggest that the website should be reviewed as well prior to publication.

10. The abstract mentions that the accession numbers for the sequences would be provided during review (which they were not). Please provide them.

Author Response

This is a descriptive manuscript focused on denovo assembled genome of the wild Caribou. The authors use a combination of short-read sequencing with Chicago libraries for linkage to create a draft assembly which is 2.2 Gbp long and consists of 4,699 scaffolds. The description of the methods is sufficient, and the assembled genome and the identified genes will be a resource to the community. However, the analyses of the genome are minimal, and leave a lot to be desired. I understand the argument that this manuscript is not intended to be focused on evolutionary  themes, so I will limit my suggestions (for the most part) to analyses that are either focused on the assembly or the use of this genome as a resource:

- Thank-you for your comments. We are indeed focussing on releasing a high-quality caribou genome as we think it will be a valuable contribution as a resource for use in research and conservation.

1. I would suggest the authors run RepeatMasker to catalog the repeats in their genome. This is a relatively straightforwards analysis, but the repeatmasked genome will facilitate several downstream analyses, adding value to this resource.

- We have added the repeat masked genome, as well as a summary table of results from RepeatMasker, to the files available on the website.

2. A genome of the reindeer has been published (https://doi.org/10.1093/gigascience/gix102) earlier, as the authors mention in their report. That genome is 2.64 Gbp in length compared to 2.2 Gbp as assembled by the authors. kmer analyses show (as per Li et al.) that the expected genome size is 2.76 Gbp. I would suggest that the authors compare their genome to the published reindeer genome and comment on the differences in the assembled genome size for the two samples.

- I e-mailed Dovetail to see if they could estimate genome size in the same way as was done for the reindeer genome (as they did the assembly they have the information needed to do the calculation). They said they would look into it but have not yet replied. I will add this in if they get back to me before publishing. 

3. The authors mention that their assembled genome is more contiguous compared to the earlier published genome. Can they comment on whether the Chicago libraries are the reason for that, or is it that their assemblers do a better job compared to SOAPdenovo which was used in the other assembly?

- Yes, we have added a comment and citation into the text explaining that the use of Chicago libraries likely lead to our genome being more contiguous due to improved scaffolding.

4. Figure 1 does not add any value to the manuscript in my opinion. I understand that comparison to other assembled genomes should be done, but the description of the analysis is sufficient. The figure should be removed.

- We discussed it and we feel that the figure is better in the main text and would prefer to keep it. However, we will move it to the supplementary materials if that is preferred by the editor. We have added a figure to the supplementary materials showing the alignment between the reindeer genome and the bovine genome which we did for these revisions, to highlight the difference in the contiguousness of the genomes. We feel that having the comparison between the two is useful.

5. On the other hand, I do not understand the Supplementary file which is provided. The information I would like to see is the intervals in the assembled genome along with the corresponding coordinates in the bovine genome that they align to. I would suggest using a BEDPE sort of format which will be useful.

- We have changed the supplementary file into an excel spreadsheet with better headings to make it more easily understandable. This outlines on which bovine chromosomes the different caribou scaffolds are located and so we think it is useful information. We have also now added a bam alignment of the two, and a BED file of the alignment (BEDPE is not possible with the fasta to fasta alignment that we did), which have detailed co-ordinates of the alignments.

6. Another relatively straightforward analysis that seems to be missing is the analysis of the mitochondrion. Does the assembly include the mitochondrion. Li et al. used the reindeer genes to calculate the phylogenetic relationships to the other genomes. Is that relationship supported by the mtDNA analysis? 

- The mitochondrial reads were excluded from this assembly. We have developed long-range PCR primers and sequenced multiple caribou mitogenomes which will also be published soon and made available (along with the primer sequences). We feel that, as you say, the focus of this manuscript is to make the caribou genome available as a resource for research, and the phylogenetic relationships demonstrated in Li et al. are well established.

7. How many of the genes identified by the authors are protein-coding. And can they enumerate the other classes of genes? How many rRNA, miRNA, tRNA genes were identified?

- We have clarified that the number of protein-coding genes is 33,177. Augustus does not identify rRNA, miRNA or tRNA genes. 

8. Kharzinova et al. (https://journals.plos.org/plosone/article?id=10.1371/journal.pone.0207944) published an analysis of the genetic diversity and population structure of domestic and wild reindeer. If you project the SNPs of the individual used in this analysis on to the MDS plot of the samples used in Kharzinova et al., how does it compare?

- As our genome is likely diverged from the reindeer used in the above-mentioned study as they are from Russian populations, and so our genome would probably sit very separate from the other points on their plot. In addition, to do a meaningful comparison, we would have to download their data, locate the same ~8,000 SNPs from our genome and extract them, and then re-do their analysis which would take longer than 10 days given for the revisions. However, we have included the citation in our manuscript as the comparison between wild and captive reindeer is interesting.

9. The authors mention that the site www.caribougenome.ca would go live by the time of the publication. Considering that this manuscript is primarily describing a resource, I would suggest that the website should be reviewed as well prior to publication.

- The website it now live and so it can be reviewed.

10. The abstract mentions that the accession numbers for the sequences would be provided during review (which they were not). Please provide them.

-We have now added the BioProject accession number for the reads (PRJNA549927) into the manuscript.

Round  2

Reviewer 3 Report

Please see inline comments to some of the replies from the authors.

2. A genome of the reindeer has been published (https://doi.org/10.1093/gigascience/gix102) earlier, as the authors mention in their report. That genome is 2.64 Gbp in length compared to 2.2 Gbp as assembled by the authors. kmer analyses show (as per Li et al.) that the expected genome size is 2.76 Gbp. I would suggest that the authors compare their genome to the published reindeer genome and comment on the differences in the assembled genome size for the two samples.

- I e-mailed Dovetail to see if they could estimate genome size in the same way as was done for the reindeer genome (as they did the assembly they have the information needed to do the calculation). They said they would look into it but have not yet replied. I will add this in if they get back to me before publishing. 

I am not too concerned about the estimation of the genome size. I am however concerned about the completeness of the genome from the authors. I would like them to compare their genomes to the genome in https://doi.org/10.1093/gigascience/gix102 and comment on the differences, specifically as they pertain to the 0.44 Gbp. Since this is a manuscript that focuses on the assembled genome, it is important to know if this is better than the other published assembly from the same species.

3. The authors mention that their assembled genome is more contiguous compared to the earlier published genome. Can they comment on whether the Chicago libraries are the reason for that, or is it that their assemblers do a better job compared to SOAPdenovo which was used in the other assembly?

- Yes, we have added a comment and citation into the text explaining that the use of Chicago libraries likely lead to our genome being more contiguous due to improved scaffolding.

This is an unsatisfying reply. What sort of analysis did the authors do that supports their claim of "likely"?

6. Another relatively straightforward analysis that seems to be missing is the analysis of the mitochondrion. Does the assembly include the mitochondrion. Li et al. used the reindeer genes to calculate the phylogenetic relationships to the other genomes. Is that relationship supported by the mtDNA analysis? 

- The mitochondrial reads were excluded from this assembly. We have developed long-range PCR primers and sequenced multiple caribou mitogenomes which will also be published soon and made available (along with the primer sequences). We feel that, as you say, the focus of this manuscript is to make the caribou genome available as a resource for research, and the phylogenetic relationships demonstrated in Li et al. are well established.

Can this be explicitly stated in the manuscript, that the mtDNA is not included in the assembly? What would happen if someone blast'ed reads from the mtDNA on the website?

7. How many of the genes identified by the authors are protein-coding. And can they enumerate the other classes of genes? How many rRNA, miRNA, tRNA genes were identified?

- We have clarified that the number of protein-coding genes is 33,177. Augustus does not identify rRNA, miRNA or tRNA genes. 

Again, this is significantly higher that the 21,155 protein coding genes identified in https://doi.org/10.1093/gigascience/gix102. What is the reason for those differences?

Author Response

I am not too concerned about the estimation of the genome size. I am however concerned about the completeness of the genome from the authors. I would like them to compare their genomes to the genome in https://doi.org/10.1093/gigascience/gix102 and comment on the differences, specifically as they pertain to the 0.44 Gbp. Since this is a manuscript that focuses on the assembled genome, it is important to know if this is better than the other published assembly from the same species.

 - We have added in text about the difference in genome size, and how the reindeer may cover more of the complete genome. However, we also ran QUAST analyses for both genomes compared to the bovine reference genome and realised that our genome as a much lower level of missing data (0.708% vs 3.597% for the reindeer). We also used QUAST to directly compare the number of genes recovered from the bovine annotation, in order to do a direct comparison between the two. We find that we can recover 8,402 of the genes, whereas the reindeer assembly only recovered 5,755. This indicates that our assembly, even if smaller, is of a higher quality.

We have also added a statement highlighting that, as we say in text, caribou and reindeer vary genetically and given that our genomes are from different continents, they likely both represent important but different genomic variation. In addition, we note that captive and wild populations have been shown to have genetic differences, and as the reindeer is a captive individual and ours a wild one – these could both be useful for different reasons. We think that, as one of our primary aims is the conservation of wild populations, this genome is needed as a valuable resource and we are keen to make it publicly available for this purpose.

This is an unsatisfying reply. What sort of analysis did the authors do that supports their claim of "likely"?

 - The assemblies have used very different methods so difficult to compare with an analysis. We talked with Dovetail and they confirmed that the Chicago method vastly improves scaffolding and is the reason why we have far fewer, far longer scaffolds and a more contiguous assembly. We have expanded in text to give detail on exactly how, and now say that the Chicago method, which produces proximity ligation libraries which have a relationship between within-read pair distance and read count, produces long-range sequence scaffolds during the assembly of genomes and is the reason for our more contiguous assembly.

Can this be explicitly stated in the manuscript, that the mtDNA is not included in the assembly? What would happen if someone blast'ed reads from the mtDNA on the website?

 - We have added a line into the methods explaining that the assembly does not include the mtDNA, but that we aim to release full mitogenomes in future. We have also added text to the BLAST page on the website to say the same thing.

 Again, this is significantly higher that the 21,155 protein coding genes identified in https://doi.org/10.1093/gigascience/gix102. What is the reason for those differences?

 - We have added into the text a statement about the difference in the number of recovered genes. We state that, this could be because our analysis does not take into account pseudogenes or incomplete proteins, but that this could also be due to the fact that their genome is a lot more fragmented with a lot more missing data. This could have impacted the detection of genes. The fact that we recovered more of the BUSCO genes, and our QUAST results (as we state above), supports this idea.